# Bioinformatic Analysis of Codon Usage Bias of HSP20 Genes in Four Cruciferous Species

**DOI:** 10.3390/plants13040468

**Published:** 2024-02-06

**Authors:** Huiyue Ji, Junnan Liu, Yineng Chen, Xinyi Yu, Chenlu Luo, Luxi Sang, Jiayu Zhou, Hai Liao

**Affiliations:** School of Life Science and Engineering, Southwest Jiaotong University, Chengdu 610031, China; jhywork123@163.com (H.J.); liujunnan2001623@icloud.com (J.L.); chenyineng1028@163.com (Y.C.); yuxinyi205@163.com (X.Y.); chenluuu525@163.com (C.L.); sangluxi8671@163.com (L.S.)

**Keywords:** Brassicaceae, codon usage bias, *HSP20* genes, mutation pressure, natural selection

## Abstract

Heat shock protein 20 (HSP20) serves as a chaperone and plays roles in numerous biological processes, but the codon usage bias (CUB) of its genes has remained unexplored. This study identified 140 *HSP20* genes from four cruciferous species, *Arabidopsis thaliana*, *Brassica napus*, *Brassica rapa*, and *Camelina sativa*, that were identified from the Ensembl plants database, and we subsequently investigated their CUB. As a result, the base composition analysis revealed that the overall GC content of *HSP20* genes was below 50%. The overall GC content significantly correlated with the constituents at three codon positions, implying that both mutation pressure and natural selection might contribute to the CUB. The relatively high ENc values suggested that the CUB of the *HSP20* genes in four cruciferous species was relatively weak. Subsequently, ENc exhibited a negative correlation with gene expression levels. Analyses, including ENc-plot analysis, neutral analysis, and PR2 bias, revealed that natural selection mainly shaped the CUB patterns of *HSP20* genes in these species. In addition, a total of 12 optimal codons (ΔRSCU > 0.08 and RSCU > 1) were identified across the four species. A neighbor-joining phylogenetic analysis based on coding sequences (CDS) showed that the 140 *HSP20* genes were strictly and distinctly clustered into 12 subfamilies. Principal component analysis and cluster analysis based on relative synonymous codon usage (RSCU) values supported the fact that the CUB pattern was consistent with the genetic relationship at the gene level and (or) species levels. These results will not only enrich the *HSP20* gene resource but also advance our understanding of the CUB of *HSP20* genes, which may underlie the theoretical basis for exploration of their genetic and evolutionary pattern.

## 1. Introduction

The genetic code is the coding information of mRNA and uses 61 codons to encode the 20 amino acids, in addition to three stop codons UAG, UGA, and UAA. Among the 20 amino acids, Met and Trp are encoded by the single codons AUG and UGG, respectively, while the remaining 18 amino acids are encoded by at least two codons, resulting in several synonymous codons. However, the choice of synonymous codons is not, at present, at uniform frequencies in the gene transcript to encode all amino acids, a phenomenon called codon usage bias (CUB) [1]. Prior researchers have identified a variety of factors contributing to the CUB patterns, including natural selection, mutation pressure, genomic structure, gene length, structure of anticodon binding domain in tRNA, and gene expressional levels [2,3]. To date, the usage frequency of synonymous codons has been revealed to be unequal across various species, between nuclei and organelle genes, and even within genes [4]. Notably, it was evident that genes in monocots and dicots exhibited a significant difference in synonymous codon usage [5] characterized by higher AT content in dicots and higher GC content in monocots, implying a connection between CUB and the evolutionary history of organisms [6]. Furthermore, CUB analysis has shed light on the evolutionary adaptation of virus to plant host [7] and the domestication of cultivated crops [8]. CUB was also a predictor of gene expression levels, as a high CUB was concluded to be associated with high gene expression levels [9,10]. For instance, CUB has been utilized in developing a live-cell monitoring system to evaluate the bioavailability of compounds in cyanobacteria, since the cells containing preferred-codon genes showed more robust growth over those containing rare-codon genes [11]. In addition, the high CUB is conducive to improving the stability of mRNA [12], aiding in protein function prediction [13], and enhancing the global translation rates [14]. An approach involving the substitution of rare codons with preferred synonymous codons, known as codon optimization, is a promising method for boosting target protein production, especially in heterologous hosts with a different CUB [14]. At present, CUB analyses based on plant genomes are highly available due to the extensive sequencing of nuclei and organellar genomes, leading to valid and credible results [4,15]. Accumulating advances based on the genomes revealed that several factors, in particular natural selection, played a significant role in CUB [10,16,17]. Conversely, the significance of CUB in gene families has not been well studied. A gene family, composed of genes derived from common ancestral genes, shows diverse functions in the regulation of plant growth and development, the participation of hormone signal transduction, and the response to biotic and abiotic stresses [18]. Therefore, it will be of great significance to investigate the CUB of gene families by which their evolutionary history and complex expressional patterns might be understood. 

Among plant gene families, heat shock protein 20 (HSP20) is particularly attractive for its ubiquitous presence in both prokaryotes and eukaryotes [19]. HSP20 acts as a chaperone and is involved not only in plant morphogenesis and physiological growth but also in responses to various environmental stresses [20]. These proteins are relatively small, with a molecular weight spanning from 14 to 42 kDa. HSP20 is composed of three domains: an N-terminal domain that binds to the target protein, an intermediate *α*-crystallin domain that enhances the stability of the target protein, and a short C-terminal extension domain facilitating homologous oligomerization [21]. In accordance with its subcellular localization, sequence identity, and phylogenetic relationship, HSP20 can be divided into at least 17 subfamilies [22]. Subfamilies CI-CXII, MI, MII, P, ER, and Px are localized to the nucleus/cytoplasm, mitochondria, mitochondria, plastids, endoplasmic reticulum, and peroxisomes, respectively. To date, the reported HSP20 families from *Coix lacryma*-*jobi* [23], *Cucumis sativus* [22], *Glycine max* [24], and *Solanum lycopersicum* [25] have been reported to consist of 32, 30, 51, and 42 putative members, respectively. Phylogenetic analyses revealed that HSP20 in the same species was divided into different clades, implying that the duplication that led to these different clades occurred before the divergence of these species [24]. Selective pressure during the evolutionary process might be a driving factor influencing the expression abundance of HSP20 [26]. Furthermore, the expression patterns of various HSP20 members were diversified under different conditions, such as drought, salinity, and heat [25]. More importantly, heterologous expression of *LimHSP16.45* can enhance the cell viability of the latter under high temperature, high salt, and oxidative stress [27], while the overexpression of *AsHSP17* increases salt sensitivity during germination and post-germinative growth [28]. Given that HSP20 confers resistance to various environmental stresses, improving the expression efficiency of *HSP20* genes using codon optimization might be an important method to develop crops that are resistant to adverse environmental stresses. Overall, these traits made the *HSP20* gene family a model for CUB research. 

The *Brassicaceae* family, encompassing a wide variety of vegetable and ornamental plants, is rich in genetic diversity, with significant potential for improving food crops. Specifically, *Brassica napus* and *Brassica juncea*, used as vegetable foods for millions of people, are rich in carotenoids, nutrients, and vitamins [29]. *Camelina sativa*, known for its strong tolerance to environmental stress and easy cultivation, presents many beneficial properties [30,31]. *Arabidopsis thaliana*, acting as a model plant, has been widely used in basic botanical research [32]. Currently, several efforts are focused on the development of *Brassicaceae* varieties that endure biotic and abiotic stresses but possess higher yields. In this study, *HSP20* genes in the nuclei genomes of *Arabidopsis thaliana*, *Brassica napus*, *Brassica rapa*, and *Camelina sativa* were wholly identified. Several CUB parameters, including ENc, RSCU, CAI, CBI, and Fop, were estimated to understand the CUB patterns of *HSP20* genes in different cruciferous species [33], which is different from previous studies. To our knowledge, this is the first study to explore the factors shaping the CUB of *HSP20* genes in various cruciferous species and might provide information for improving genetic breeding on the basis of *HSP20* genes.

## 2. Results

### 2.1. Identification and Nucleotide Composition of HSP20 Genes

A total of 140 nonredundant *Hsp20* genes were identified from four cruciferous species, comprising 27 genes from *A. thaliana*, 48 genes from *B. napus*, 36 genes from *B. rapa*, and 29 genes from *C. sativa* (Appendix A). The CDS lengths of the *HSP20* genes ranged from 393 bp (*Csa19g056510*) to 1116 bp (*Csa20g006100*), with a maximum difference of 723 bp, and encoded proteins ranging from 130 to 371 amino acids in length. In addition, the predicted isoelectric point (pI) of HSP20 varied from 4.37 (*Csa19g056510*) to 9.78 (*CDY70225*), with 44.29% having a pI value greater than 7, indicating that more than half of the HSP20 proteins were neutral or acidic.

The A base is the richest base in the 140 *HSP20* genes, with an average content of 29.39%, followed by the G base, which averaged 26.60%. The average GC content of the 140 *HSP20* genes is 47.4% and ranged from 36.2% to 58.2%, with the lowest GC content contained in *CDY45944* and *Bra008274* and the highest in *Bra024931*. In terms of species, *HSP20* genes in *B. napus* have the highest GC content (48.1%), while those in *A. thaliana* have the lowest (46.5%). 

### 2.2. Codon Usage Indices of HSP20 Genes 

As shown in Figure 1, the average GC1, GC2, and GC3 contents of 140 *HSP20* genes from four species were 53.06%, 38.69% and 50.20%, respectively. The GC1 content across the four species varied from 42.31% to 60.24%, while the GC2 content of the four species varied from 24.61% to 48.28%. The average GC3s value of 48.5% indicated that the *HSP20* genes preferred an A or T base as the third base. Furthermore, the indices of A3s, T3s, G3s, and C3s indicated that *HSP20* genes tended to end with a T base, followed by the G, C, and A bases, with respective values of 35.2%, 33.1%, 29.3%, and 29.2%. The CAI values in 140 *HSP20* genes ranged from 0.154 (*CDX79100*) to 0.318 (*Bra024931*), with a mean value of 0.225. As the expression level of genes and preference often showed a positive correlation, the CAI value of *HSP20* genes indicated the relatively low expression levels of *HSP20* genes. Among the four species, the *HSP20* genes in *C. sativa* had the highest CAI values, followed by those of *A. thaliana*, *B. napus*, and *B. rapa*, implying that those in *C. sativa* had relatively high expression levels. The CBI values of global *HSP20* genes ranged from −0.177 (*CDY70473*) to 0.304 (*Csa11g019390*), with a mean CBI value of 0.033. Among the four species, *A. thaliana* had the lowest CBI value of 0.018, while *C. sativa* had the highest CBI value of 0.048. With respect to Fop, its ranging region was from 0.327 (*Csa05g059340*) to 0.591 (*Csa11g019390*), with an average of 0.437. In terms of species, *C. sativa* showed the highest Fop value (0.446), whereas *A. thaliana* showed the lowest (0.429). Additionally, the ENc values were calculated to be from 37.900 (*Bra018383*) to 61.000 (*Bra022078*, *Bra028253*, and *CDY00476*), with an average of 51.814. Regarding the four species, *HSP20* genes in *B. rapa* were considered to have the strongest CUB due to the lowest ENc value of 51.344, while those in *B. napus* showed the weakest CUB due to the highest ENc value of 52.310. The negative GRAVY values of *HSP20* genes in the four species confirmed their hydrophilic nature, suggesting their unlike location in membrane structure. The aromaticity (AROMO) values, representing the usage frequency of Phe, Tyr, and Trp residues, varied from 0.033 to 0.136, with a mean value of 0.069, confirming that the composition of aromatic amino acid residues varied little. 

### 2.3. ENc-GC3s Plot Analysis

To reflect the correlation between the ENc and GC3s values, a correlation curve was generated. According to Appendix A, the ENc values of *HSP20* genes varied widely among genes, with an average of 51.814, indicating a generally low CUB of *HSP20* genes in the four studied species. Statistically, no *HSP20* gene had low ENc values (≤35) in the four species with a threshold value of strong bias, while 94 genes had high ENc values (50–61) with weak bias. Given that strong- and weak-bias genes were 0 and 67.14%, respectively, in the four species, the CUB of *HSP20* genes was determined to be relatively low. 

As shown in Figure 2, most of the genes resided below the expected curve (blackline), except that a few genes of *B. napus* and *B. rapa* were situated above and (or) on the expected curve. The expected curve denotes the expected position of genes where the CUB was governed solely by mutation pressure, while the CUB of those genes distributed below the expected curve was mainly determined by natural selection. Consequently, natural selection was inferred as the primary influencer of the CUB pattern of *HSP20* genes in this study, a finding reflected in similar analyses of apple, *B. napus*, and citrus [4,34,35].

### 2.4. PR2-Plot Analysis

Parity rule 2 (PR2) plots were conducted to ascertain any inequality between pyrimidines and purines at the third position (Figure 3). The plot, centered on 0.5, was divided into four quadrants. In case the mutation pressure alone affects codon bias, the occurrence of A/T or C/G at codon position three is equal, while natural selection pressure leads to the unequal usage of A/T or C/G at the third position [36]. As shown in Figure 3, the *HSP20* genes of the four cruciferous species were not evenly distributed in the four regions, indicating that strong bias played a role in the composition of the third codon position. Over half of the genes were located in the lower region (A3/(A3 + T3) < 0.5), revealing the preponderance of T over A. Similarly, more than half of the genes were located in the right region (G3/(G3 + C3) > 0.5), demonstrating that the occurrence of G in the third position was more frequent than that of C. Interestingly, a few dots in the upper left quadrant suggested a distant codon usage preference in *HSP20* genes.

### 2.5. Neutrality Plot

The relationships between GC12 and GC3 in the four species were separately examined using a neutrality plot (Figure 4). The content of GC12 ranged from 36.86% to 52.64%, while that of GC3 ranged from 33.74% to 77.33%. A weak positive correlation between GC12 and GC3 was observed in the four species, with *R* values ranging from 0.197 to 0.282, suggesting that CUB was affected by both natural selection and mutation pressure. Additionally, the contributing extents of mutation pressure and natural selection on CUB can be evaluated using a neutrality plot [2,15]. As a result, the small regression coefficients, varying from 0.0522 to 0.1034, indicated that mutation pressure contributed less than 10.34%. Conversely, the remaining 89.66% of the bias was attributed to natural selection. Therefore, these results strongly underscored the dominant role of natural selection in the CUB pattern, but mutation pressure plays a minor role in determining the CUB of *HSP20* genes in the four cruciferous species. Similar results have been noted in seven Rosales species [37].

### 2.6. Correlation Coefficient Analysis of Codon Usage Indices

It was observed that the CBI showed a highly significant correlation with the CAI and Fop indices in all species (*p* < 0.001) (Figure 5). Likewise, C3s showed a highly significant correlation with FOP, CBI, and CAI (*p* < 0.001), except for the C3s and CAI in *C. sativa* (*p* < 0.05). Intriguingly, a significant correlation between the GC content and GC3 in all species was evident (*p* < 0.001), indicating that the GC composition of *HSP20* genes could markedly affect GC3 content substantially. GC content also exhibited a correlation with GC1 and GC2 content in *A. thaliana, B. napus*, *B. rapa*, and *C. sativa*, confirming that the GC composition also could affect the GC1 and GC2 content to some extent (Figure 5A–D). Meanwhile, the GC content exhibited a highly significant correlation with T3s, C3s, and A3s across all species (*p* < 0.001). Except for the G3s in *A. thaliana* and *C. sativa*, those in the other two species correlated with the GC content (*p* < 0.05), implying that the GC composition impacted the base at the third position (Figure 5). We also found that GC3, A3s, T3s, C3s, and G3s were extremely significant in the four species (*p* < 0.001).

### 2.7. Relationship between ENc and Gene Expression Level

FPKM is an indicator of the gene expression level, with a lower value denoting a lower gene expression level [38]. The FPKM values of 27 *HSP20* genes in *A. thaliana* revealed that the expression pattern of the *HSP20* genes was complex and varied under various conditions (Appendix A). Some *HSP20* genes, like *AT5G12030*, were notably induced under drought and salinity stress compared with the control condition. We performed a Karl Pearson’s correlation analysis of FPKM and ENc, and a negative correlation that ranged from −0.0320 to −3.3535 was observed, indicating that gene expression levels might be affected by the CUB (Figure 6).

### 2.8. Identification of Optimal Codons

The RSCU values for the *HSP20* genes in the four species were depicted in a stacked bar chart (Figure 7), with detailed RSCU values listed in Appendix A. These stacked bars demonstrated high genetic conservation across species. After the removal of stop codons and codons with RSCU = 1, there were 29 (*A. thaliana)*, 28 (*B. napus*), 27 (*B. rapa*), and 26 (*C. sativa*) high-frequency codons with RSCU over 1 in the four cruciferous species. A total of 12 codons satisfying RSCU > 1 and ΔRSCU ≥ 0.08 were grouped as optimal codons, of which 6, 6, 3, and 1 optimal codons were detected in *A. thaliana*, *C. sativa*, *B. rapa*, and *B. napus*, respectively. UAC was shared by *B. napus* and *B. rapa*. AAC and GGU were shared by *A. thaliana* and *C. sativa*, while AGA was shared by *B. rapa* and *C. sativa*. The remaining codons AGG, CCA, GAA, GAG, GCU, GUG, UCU, and UGA were used as the optimal codons for the individual species. The analysis revealed a preference for codons ending with A/T, as evidenced by 4 of 12 optimal codons ending with A, 3 each with T and G, and only 2 with C. 

### 2.9. Comparison of Evolutionary Relationship and Cluster Analysis

The evolutionary relationship of *HSP20* genes was constructed on the basis of their CDSs using the NJ method. This resulting phylogenetic tree was highly supported, with more than 75% nodes obtained bootstrap over 90 BP (Figure 8). In the phylogenetic tree, the *HSP20* genes were clustered into various clades, reflecting their relationship. Genes in the same clades revealed a close relationship, while those in different clades were distant during evolution. As shown in Figure 8, 140 *HSP20* genes were classified into 13 subfamilies, including CI, CII, CIII, CV, CVI, CVII, CIV, ER, MI, MII, P, Po, and HSP20-like, containing 13, 3, 4, 2, 3, 9, 6, 3, 4, 4, 4, and 85 CDSs, respectively. Notably, our phylogenetic results indicate that *HSP20* genes from the same species were separated into different clades, suggesting that gene duplication leading to these different clades occurred before the species divergence. For instance, 5, 4, 2, and 1 *HSP20* genes from *A. thaliana*, *C. sativa*, *B. rapa*, and *B. napus*, respectively, were included in subfamily CI, similar to the phylogenetic pattern observed in the *expansin* genes in the *Brassica* species [34].

Meanwhile, cluster analysis based on the indices provides statistical visualization to compare the codon usage similarity and has been widely used in plant CUB analysis [37,39]. The indices employed in previous studies included GC3 in *Ginkgo biloba* [39] and RSCU values in *Rosales* species [37]. In our study, RSCU values were used to construct the cluster tree, and the *HSP20* genes were grouped into various subfamilies rather than by species (Appendix A), similar to the results in phylogenetic tree. The *HSP20* genes in the same subfamily had a close distance, and those with close genetic relationship shared a similar CUB. Furthermore, another cluster tree was created using four super-HSP20 genes, each representing a composite of the *HSP20* genes from the same species (Appendix A). In this tree, *B. napus* and *B. rapa* formed a monophyletic group (Appendix A), indicating that closer genetic relationships resulted in more similar CUB patterns among the species [4]. 

### 2.10. PCA of RSCU Values with Species and Subfamilies

In the phylogenetic tree, the clustering manner of *HSP20* genes was dependent on functional subfamilies rather than species (Figure 8). Genes within the same functional subfamilies exhibited close relationships, irrespective of species origin. We performed PCA to explore whether the CUB pattern of the *HSP20* genes was involved with evolutionary history, and hence, the PCA plots of the first and second principal components were generated against each other in terms of subfamilies and species, respectively. As depicted in Figure 9A, we observed that all dots divided by different species were dispersed with no clear clustering trend. However, in Figure 9B, the genes within the same functional subfamilies were close to each other, and the majority of genes from the 12 subfamilies, excluding those in the *HSP20*-like subfamilies, were clustered on the basis of subfamilies. The results strongly indicated that the CUB pattern of the *HSP20* genes was intensively linked to their evolutionary history, with genes that shared a close evolutionary relationship exhibiting similar codon usage bias.

## 3. Discussion

The synonymy of the genetic code decreases the mutation of protein products, even though variation in the genetic code occurs within a gene [2]. Nevertheless, the CUB occurs ubiquitously and contributes to numerous biological processes, including the intensive production of exogenous proteins, RNA stability, environmental adaptability, viral infection, and evolutionary history [4,12,40,41]. Moreover, synonymous codons were believed to affect protein folding due to a translational pause caused by rare codons, and hence, the codon-specific structural properties were conducive to offering insights into the potential causes of conformational diseases [14]. Among the various biological factors shaping the CUB, mutation pressure and natural selection are probably major constituents of the evolutionary forces on the CUB in which natural selection would be more preponderant in the high-expression genes than in the low-expression genes on the condition that natural selection outweighs mutation pressure [42,43]. Given the continuous release of greenhouse gases [44], plants are expected to encounter various environmental stresses, especially temperature, drought, and oxidative stress. The *HSP20* genes are instrumental in plant responses to different environmental stresses, such as high temperature, drought, cold, and oxidative stress, because they can modulate their mRNA and (or) protein levels [21,38], and thus, it is attractive to carry out CUB analysis on *HSP20* genes to evaluate whether mutation pressure or natural selection exerts more influence on the CUB pattern. Nowadays, several CUB analyses are achieved at the genomic level due to the completion of plant genomes [4], which showed that the CUB pattern was associated with a genetic relationship. However, studies focused on the CUB of gene families remained scarce, which led to an insufficient understanding of CUB patterns. 

In the present study, genome-wide mining was implemented, resulting in the identification of 140 *HSP20* genes in four species. The number of *HSP20* genes in four species is more than in fungi, algae, bryophytes, pteridophytes, and gymnosperms [45]. It was suggested that the *HSP20* genes experienced an initial expansion during the landing process of algae, followed by further expansion in seed plants and angiosperms [45]. On certain conditions, this greater expansion of the gene family was attributed to whole genome duplications, as seen in soybean [24]. Previous phylogenetic analyses succeeded in the classification of the *HSP20* family into distinct subfamilies, but their number varied across the plant lineage [24]. Comparative analyses of the genomes of *Solanum lycopersicum*, *Cucumis sativus*, and *Coix lacryma-jobi* identified 42, 30, and 32 *HSP20* family members, classified into 13, 11, and 13 subfamilies, respectively [22,23,25]. In the present study, the *HSP20* genes from different species that clustered into the same subfamilies showed closer relationships than those from the same species but different subfamilies. These findings suggest that the *HSP*20 genes in the four cruciferous species likely originated from a common ancestor before the divergence of these species, leading to the different subfamilies. Therefore, the second issue to be addressed is whether the CUB pattern of *HSP20* genes in the same subfamily is closer than that in the same species.

The mean GC content of the *HSP20* genes in these species ranged from 46.5% to 48.1%, indicating that their CDSs were AT-rich. The GC content varied during the evolutionary process, with the highest in algae and the lowest in dicots [4]. In the dinoflagellate *Alexandrium tamarense* CCMP 1598, the average GC content was 58.4% [2,46]. In addition, the GC contents of nuclei genes were higher than those of organelle genes [47]. Among GC1, GC2, and GC3 values, the highest content of GC1 and the lowest content of GC2 were obtained. Previous studies in *A. tamarense*, *Camellia sinensis* cultivars, and *Triticum aestivum* also exhibited the lowest GC2 content [2,47,48]. On the contrary, the lowest GC3 content was found in 18 *Oryza* species [2], implying the diversity of base composition in different plants. The *HSP20* genes preferred A- and T-ending codons due to the global GC3s percentage of 48.5%, consistent with previous studies in which dicot genes tend to end with A or T, while monocot genes preferred G- or C-ending codons [49]. Furthermore, the higher prevalence of T- and G-ending codons in the *HSP20* genes highlighted the inequality of bases in the degenerate codon position, facilitating the comparative analysis of CUB in the genus Bungarus [50]. Intriguingly, a significant correlation was found between overall GC content and the GC composition at three codon positions, suggesting that the CUB of the *HSP20* genes in these species was affected by both natural selection and mutational pressure, coinciding with the result of Li et al. [51] in *Amanita* species. Additionally, the majority of optimal codons in *HSP20* genes ended with A/T, paralleling the pattern observed in *MYB10* genes in Populus [13].

Various *HSP20* genes in *A. thaliana* exhibited complex expression patterns under different conditions, implying that they might have diverse biological functions. For instance, *AT5G12030* appeared to enhance drought and salt tolerance in *Arabidopsis*, probably due to its chaperone activity on citrate synthase [52]. In addition, we observed a negative correlation between FPKM and ENc, indicating that genes with higher expression exhibited a more pronounced CUB than those with lower expression [10]. Chakraborty et al. [50] also observed a significant correlation between expression levels and ENc in *Bungarus* species, highlighting the influence of CUB on gene expression. The classic theory linking the gene’s expression level to its codon preference is that frequent codons are more adapted in highly expressed genes in order to match the tRNA abundances [53,54]. The relationship between CUB and tRNA abundance contributes to the accuracy and efficiency of translation. Firstly, the preferred codons are less likely to be recognized by mismatched tRNAs during translation, thereby reducing erroneous protein production [55]. Secondly, the preferred codons, aligning with abundant tRNAs, require shorter time for matching, thus improving the efficiency of ribosome utilization [56]. In summary, genes with a high CUB tend to exhibit higher expression levels than do genes with a low CUB, and the substitution of rare codons with preferred ones in the *HSP20* genes could enhance associated protein production.

In our study, the weak correlation between GC3 and GC12 indicated the greater contribution of natural selection than mutation pressure to the CUB in cruciferous species. Additionally, the dominant role of natural selection was further strengthened by the smooth slope of the regression line in the neutrality plot; ENc-GC3s plot together with PR2 bias, which agreed with the previous study on *Cuscuta australis* [10].

Based on the phylogenetic, clustering, and PCA results, our findings also supported the notion that the *HSP20* genes with a closer phylogenetic relationship exhibited a more similar CUB pattern. Although the *HSP20* genes within the same subfamilies were not derived from the same species, these members showed a close phylogenetic relationship together with close distance (Figure 7 and Figure S1). In addition, our results provided evidence that *B. napus* and *B. rapa* shared a more similar CUB pattern than *A. thaliana* and *C. sativa* (Appendix A). Therefore, our results, along with previous reports [4,57], supported that the CUB pattern was consistent with the genetic relationship at the gene level and (or) species level and provide supplementary insight into the genetic relationship derived from phylogenetic analysis [51]. 

## 4. Materials and Methods

### 4.1. Identification of HSP20 Genes

Following the methodology of Tian et al. [58], the proteomes of four cruciferous species were downloaded from the Ensembl plants database (https://plants.ensembl.org/index.html) (accessed on 1 June 2023) for use as local datasets. A Hidden Markov model (HMM) approach, employing the Pfam (http://pfam.xfam.org/) (accessed on 20 June 2023) profile of PF00011 as query, was utilized to identify HSP20 proteins. An *E*-value threshold of less than 1 × 10^−5^ was set. Subsequently, the candidates, which contained *α*-crystallin domain and had molecular weight ranging from 14–42 KD, remained. Moreover, BLASTP tool was performed to align the members from *B. napus*, *B. rapa*, and *C. sativa* with those from *A. thaliana* proteome with a criterion of *E*-value < 1 × 10^−5^. Finally, the protein sequences of the remaining HSP20 members were resubmitted once again to Pfam, SMART (https://smart.embl.de/) (accessed on 14 September 2023), and InterPro (https://www.ebi.ac.uk/interpro/) (accessed on 16 September 2023) to verify the conserved Hsp20 domain. 

### 4.2. Codon Usage Bias Analysis

Complete CDSs of *HSP20* genes were extracted using TBtools-Ⅱv2.052, ensuring that ATG acted as the start codon and that TGA, TAA, and TAG acted as the stop codons. The relative synonymous codon usage (RSCU), codon adaptation index (CAI), synonymous codon GC content, and 3rd base contents (A3s, T3s, G3s, and C3s) were determined using CodonW 1.4.2 [59]. Codon usage frequencies were calculated with the CUSP tool on the EMBOSS services (https://www.bioinformatics.nl/emboss-explorer/) (accessed on 11 October 2023), while the effective codon numbers were obtained using CHIPS, while the effective codon numbers were obtained using CHIPS.

### 4.3. ENc-GC3s Plot 

To analyze the relationship between base composition and codon bias, an ENc-GC3s plot was created using GC3s content on the x-axis and ENc values on the y-axis. The expected ENc values were determined using the following equation [51]:ENc = 2 + GC3s + 29/(GC3s^2^ + (1 − GC3s)^2^)

The ENc values range from 20 to 61, with 35 usually considered as the threshold for assessing the strength of the CUB [60]. If the points in the ENc-GC3s plot lie on or near the expected curve, the CUB of a given gene is mostly affected by mutation pressure. Conversely, the points situated below the expected curve suggest natural selection as the main contributor to the CUB of the given gene [61].

### 4.4. PR2-Plot Analysis

A PR2 Rule bias plot was generated with A3/(A3 + T3) on the x-axis and G3/(G3 + T3) on the y-axis. If the base content of one strand is A = T and G = C, it signifies that mutation pressure and natural selection would not affect both complementary DNA strands [59]. The center of the plot is set at (0.5, 0.5), where the distance and direction from the center point represent the degree and direction of PR2 bias, respectively. A longer distance away from the center equates to a stronger PR2 bias [7]. According to the proportion of four bases, the influence of base mutation on base variation could be inferred.

### 4.5. Neutrality Plot

A neutrality plot is a commonly assessed tool to determine the extent to which mutation pressure and natural selection play a role in CUB. To discern which factor was more influential, a neutrality plot was generated using Origin, with the GC3 on the x-axis against GC12 (the average value of GC1 and GC2) on the y-axis [7]. The slope of the regression line ranged from 0 to 1; 0 represents no difference in codon base usage across the 1st, 2nd, and 3rd positions of codons, while 1 means a predominant role of mutation pressure [62]. 

### 4.6. Correlation Analysis of Codon Usage Bias Indices

SPSS 26 software was utilized to analyze the significant differences in the paired CUB parameters of the *HSP20* genes. The parameters were compared in pairs, and the number of * represented the significance level, while the numerical magnitude represented the correlation strength.

### 4.7. The Relationship between CUB and Gene Expression Level

To evaluate the relationship between the CUB and the gene expression level, the FPKM (fragments per kilobase of exon model per million mapped fragments) and ENc values of the *HSP20* genes were applied as indicators of the gene expression level and the CUB, respectively. In our prior study, we obtained the transcriptome of *A. thaliana* under normal, drought, and salinity conditions [63]. Then, the analysis was conducted using SPSS software in which we plotted the ENc values on the x-axis and the FPKM values on the y-axis.

### 4.8. Determination of Optimal Codons

Optimal codons refer to those occurring with greater frequency in high-expression genes than in low-expression genes [47]. On the basis of the ENc values, the bottom and top five genes were collected and formed the high-expression and low-expression databases, respectively. Then, the optimal codon was defined using the following two criteria. First, a given codon should possess an RSCU value greater than 1 in both databases. Second, the RSCU value of a given codon in a high-expression database is 0.08 greater than that in a low-expression database [7].

### 4.9. Phylogenetic and Clustering Analysis

Phylogenetic analysis of the *HSP20* genes in the four species was constructed using their coding sequences (CDS) and the neighbor-joining (NJ) method. The NJ method employed the P-distance replacement model and included 1000 replicates, with the gaps removed [64]. The RSCU values of 59 codons (excluding the three stop codons, AUG, and UGG) of the four species were clustered using SPSS v26.0 software. The clustering approach adopts intergroup linkage, defining the distance between genes based on the square Euclidean distance of the RSCU values. 

### 4.10. Principal Component Analysis

Principal component analysis (PCA), a multivariate statistical method, was utilized to explore the trends shaping the CUB of the *HSP20* genes [15]. Specifically, the RSCU values of 59 synonymous codons were represented as a 59-dimensional vector, which was then transformed into principal components. Subsequently, the PCA plot of the first two axes was constructed, effectively encapsulating most of the principal components shaping the CUB among the genes. 

## 5. Conclusions

In this work, 140 *HSP20* genes were identified across four cruciferous species. The CUB of the *HSP20* genes was found to be relatively low. Both mutation pressure and natural selection were proposed to contribute to the CUB, with natural selection emerging as the main contributor according to various analyses. Meanwhile, the consistency between the genetic relationship and the CUB pattern provided the necessary evidence for understanding the evolutionary process and underscored the potential strategy for genetic transformation in the future.

## Figures and Tables

**Figure 1 plants-13-00468-f001:**
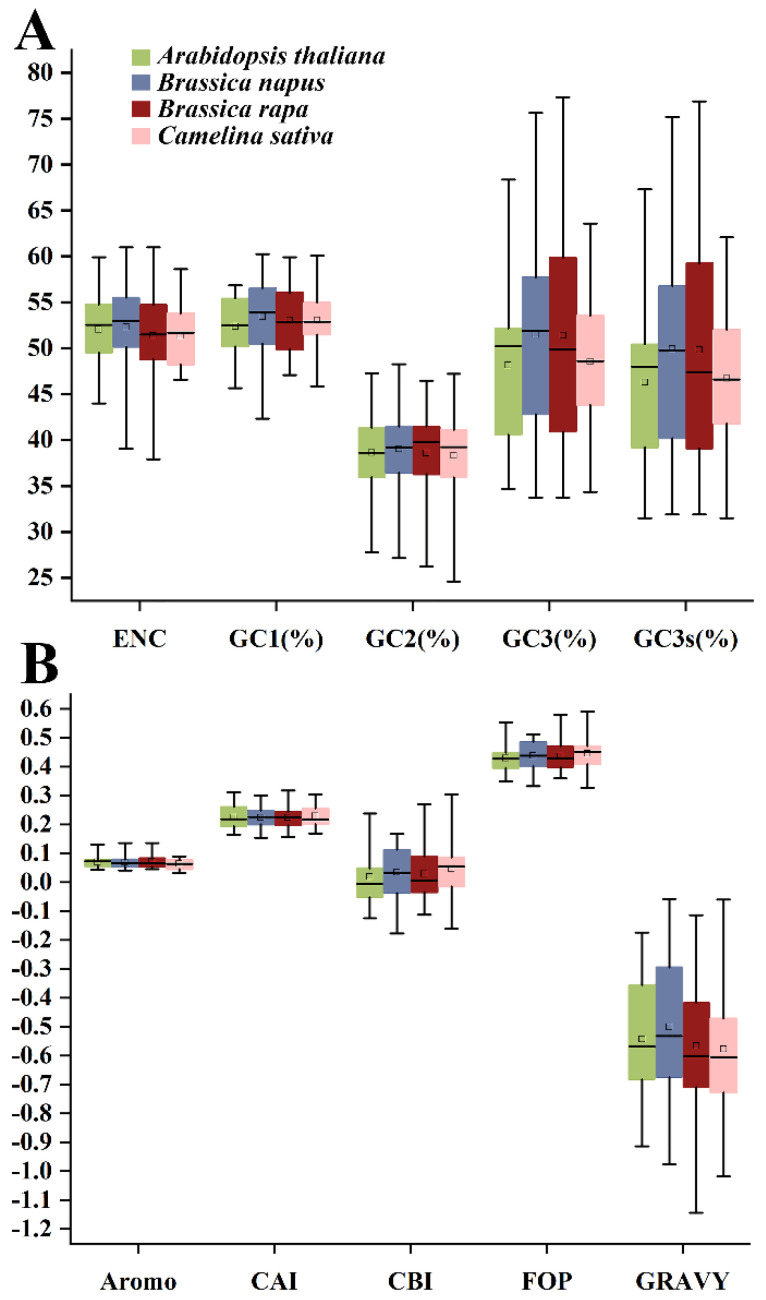
The codon usage indices of *HSP20* genes in four cruciferous species. (**A**) ENc, GC1, GC2, GC3, and GC3s indicators. (**B**) Aromo, CAI, CBI, FOP, and Gravy indicators.

**Figure 2 plants-13-00468-f002:**
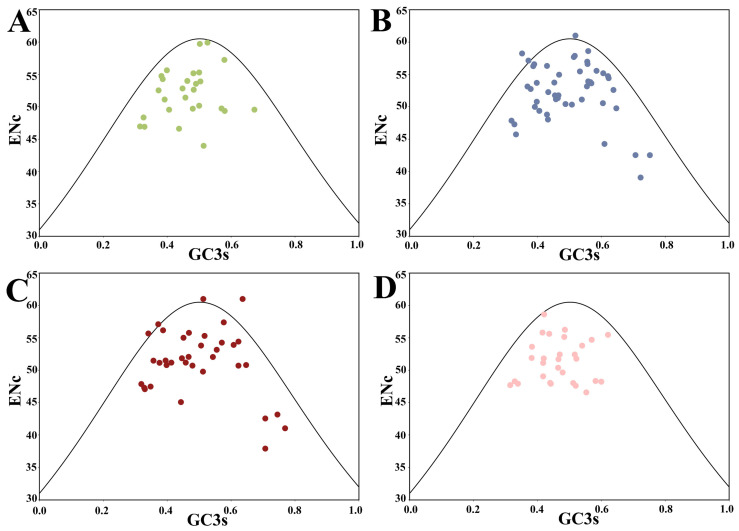
ENc-GC3s plot of *HSP20* genes in four species. The black line represents the expected curve when the codon usage bias is determined only by mutation pressure. (**A**) *A. thaliana*; (**B**) *B. napus*; (**C**) *B. rapa*; and (**D**) *C. sativa*.

**Figure 3 plants-13-00468-f003:**
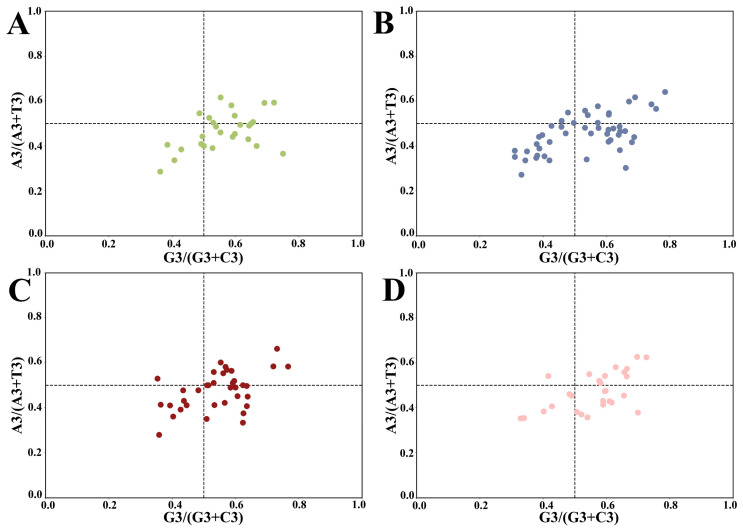
Parity rule 2 plot analysis of HSP20 genes in four species (GC bias on the x-axis and AT bias on the y-axis). The center point at 0.5 represents A = T and G = C, which means that there is no codon usage deviation between the two DNA strands. (**A**) *A. thaliana*; (**B**) *B. napus*; (**C**) *B. rapa*; and (**D**) *C. sativa*.

**Figure 4 plants-13-00468-f004:**
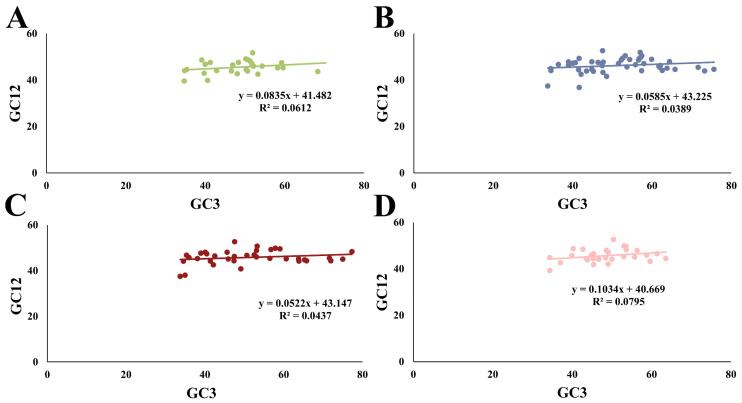
Neutrality plot between GC12 (the mean GC content at the first and second positions) and GC3 (GC content at the third codon position). (**A**) *A. thaliana*; (**B**) *B. napus*; (**C**) *B. rapa*; and (**D**) *C. sativa*.

**Figure 5 plants-13-00468-f005:**
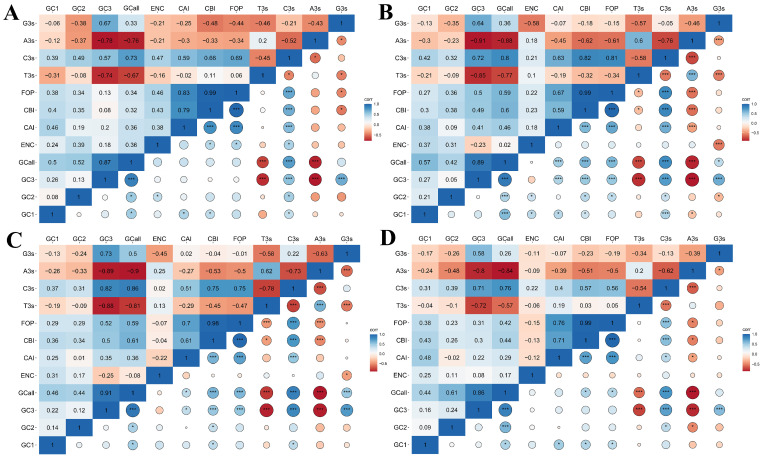
Pearson correlation analysis of codon usage indices in four cruciferous species. The color changing from red to blue represents an increasing correlation index. (**A**) *A. thaliana*; (**B**) *B. napus*; (**C**) *B. rapa*; and (**D**) *C. sativa*. * and *** represented statistical significance of *p* < 0.05 and *p* < 0.001, respectively.

**Figure 6 plants-13-00468-f006:**
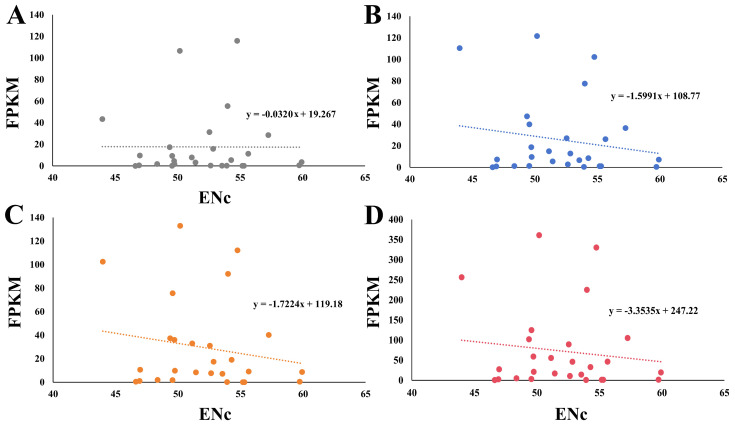
The relationship between FPKM and ENc. The FPKM values of *HSP20* genes under (**A**) normal, (**B**) salinity, and (**C**) drought conditions were applied for calculation. The FPKM values of *HSP20* genes used in (**D**) represented the sum total of those under normal, drought, and salt conditions.

**Figure 7 plants-13-00468-f007:**
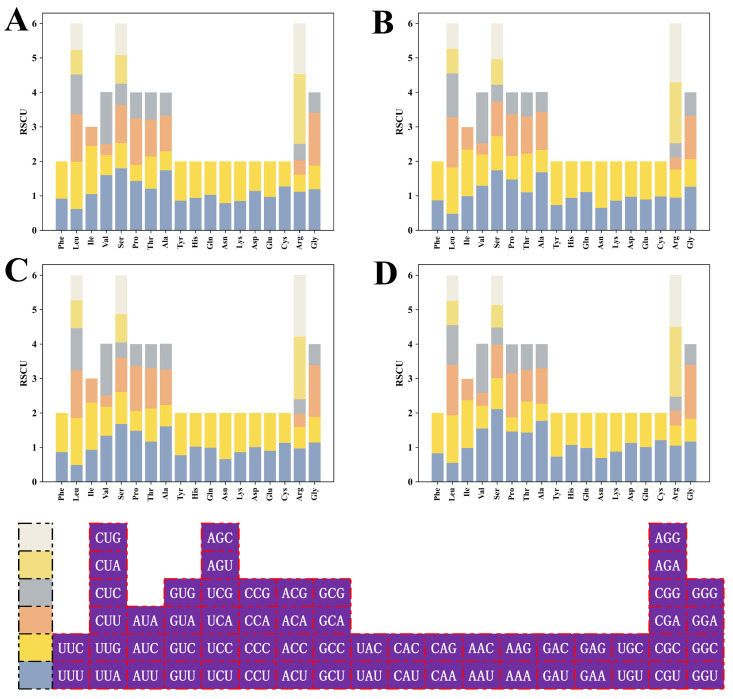
RSCU analysis of *HSP20* genes in four cruciferous species. (**A**) *A. thaliana*; (**B**) *B. napus*; (**C**) *B. rapa*; and (**D**) *C. sativa*.

**Figure 8 plants-13-00468-f008:**
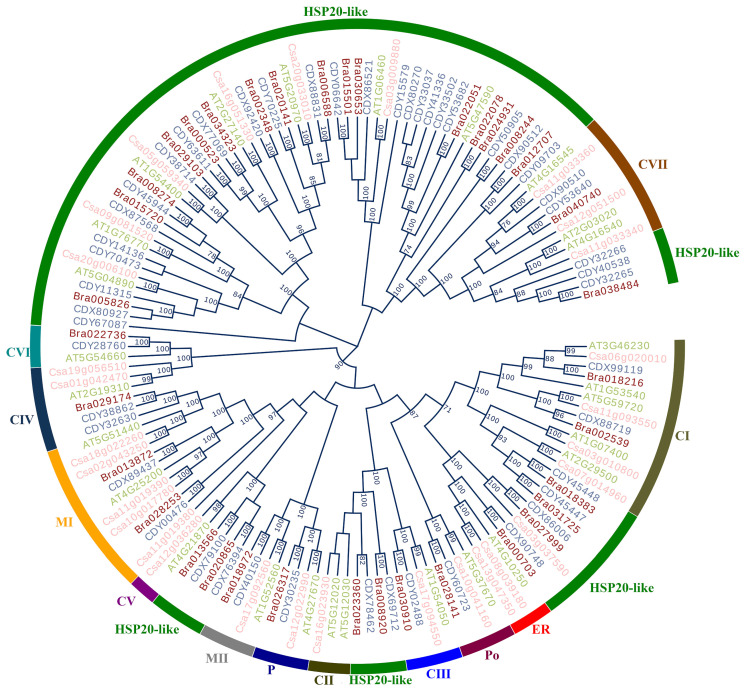
The phylogenetic analysis based on CDS of *HSP20* genes in four cruciferous species.

**Figure 9 plants-13-00468-f009:**
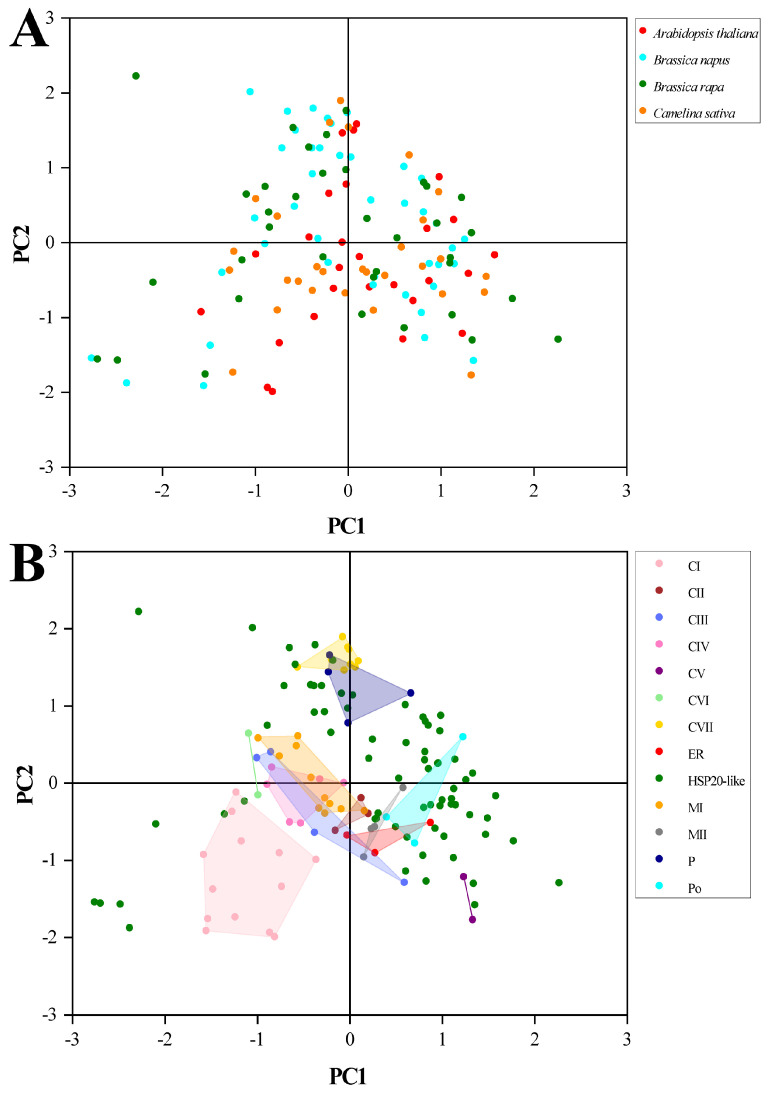
The PCA plots of first axis against second axis. (**A**) Points for *HSP20* genes classified by species. (**B**) Points for *HSP20* genes classified by subfamilies.

## Data Availability

All data are available within this publication and Appendix A.

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
