# Peer review of "Bioinformatic Analysis of Codon Usage Bias of HSP20 Genes in Four Cruciferous Species"

_plants, 2024, doi:10.3390/plants13040468_

Round 1
Reviewer 1 Report
Comments and Suggestions for Authors
Although I think the methodology is sound and the results are novel, my main doubt about the manuscript is the rationale for carrying out this work o codon usage and its significance. In order to present this the authors must greatly improve the introduction. Some points to consider explaining are:
1. Why is CUB important-does it have significance outside of evolutionary studies.
2. Include more information on CUB in general in different species and mention correlations if they exist.
3. Why is there emphasis on association with expression levels and how does this relate to evolutionary changes-what causes the changes in expression? This cannot be related to specific codons.
4. More extensive information is needed on HSP genes/HSP20 genes in general, especially in term of their evolution-what do the 12 subfamilies signify-are they related to activity or expression in Brassicas or other plant species?
5. Did the authors look at synteny to see whether the genes studied in the different Brassica species could relate to correlations in evolutionary terms.
6. Given that the results indicate the distinction between mutation and natural selection, how would domestication of cultivated species affect the correlations?
In general I think the results are well presented. Would it be useful to construct a phylogeny including only the HSP20 type genes and or include information on CUB in the dendrogram? The association between but the clades and CUB is not clear.
The discussion section is an extended description of the results-the authors must not only compare in detail their results with other reports but emphasize the similarities, differences and the significance of their findings. What does it mean if there is an association between CUB and expression for example.
Comments on the Quality of English Language
English should be revised throughout by a native English speaker-many words and phrases are out of context and cause confusion to the reader. In some sections it is difficult to understand the point being made.
Reviewer 2 Report
Comments and Suggestions for Authors
Review plants-2836940_v1
The usage of synonymous codons in the genetic code is not random. Rather, some codons encoding a given amino acid are used preferentially to encode that amino acid. This bias depends on genetic selection and mutation pressure. It is well-accepted that codon bias varies between organisms and specific genes. The codon usage is also correlated with gene expression level.
In the current manuscript, Ji et al report on an analysis of codon usage among the small heat shock proteins (HSP20) in four Cruciferous Species of commercial and nutritional importance. They conclude that natural selection rather than mutation pressure is the major force beyond the evolution of codon bias usage in these genes.
The work is well-defined and executed. However, I recommend that the comparison with other related gene families in the target species and in the general evolutionary spectrum be expanded. The manuscript presentation also needs improvement.
Major points.
1. The manuscript would be very much more interesting if the codon bias and biological function were compared with other heat shock protein families in the four target species as well as with the small heat shock protein families in other plants occupying different biological niches. I realize that this would mean extra work, but the software and databases are readily available.
2. Please discuss the relationship between codon usage and specific tRNA concentrations.
3. Discuss the potential importance of the rare codons for protein folding (https://doi.org/10.1016/j.jmb.2023.168384)
Minor points.
1. Please define the many abbreviations used in the text.
2. Line 82: I assume the authors mean “the ability of HSP20s to confer resistance to”
3. Please summarize the result listed in the text of Section 33.2 in a (Supplementary) Table
4. There is no call in the text for Figures 1 and 4
5. Table 3, line 4: frepuency>frequency.
Comments on the Quality of English Language
The English language is completely understandable, except for a few examples of unusual syntax
Reviewer 3 Report
Comments and Suggestions for Authors
General evaluation: aim, novelty, and significance,
This study investigates the codon usage bias (CUB) of the gene family of heat shock protein 20 (HSP20s), which act as chaperons and play significant roles in many biological processes. This phenomenon was studied in four cruciferous species. The results suggested that mutation pressure and natural selection might contribute to the CUB. The study clarified that the CUB pattern was consistent with the genetic relationship at the gene and (or) species levels.
The data generated by this study may contribute to an understanding of biological and molecular evolution and the intended or natural mutational changes. It will also lighten the biological changes emergent in response to different stresses.
However,
· The scope of the study should be limited to “Bioinformatic analysis”. This should be clear in the title, abstract, introduction, Materials and methods, and results and discussion.
· The resources of the bioinformatic data should also be clear in the Materials and Methods. Detailed information on the bioinformatic data should be added since the validity of the results and conclusions of this study is directly dependent on these data.
The title
The title should qualify the nature of the study as a bio-informatic one. So, it is suggested to be “Bioinformatic Analysis of Codon Usage Bias of HSP20 Genes in Four Cruciferous Species”
Abstract
§ Main
Indicate the bioinformatic approach used in this study and briefly describe the original data.
§ Minor
L18, change (showed significant correlation) into (significantly correlated).
L20, change (CUB) into (the CUB)
L21, change (exhibited negative) into (exhibited a negative)
L26, change change (CUB) into (the CUB)
L26, change (supported that CUB pattern) to (supported the fact that the CUB pattern)
L27 change (gene level) into (the gene level)
L27 change (species level) into (species levels)
L28, change (understanding on) to (understanding of)
Introduction
§ Major
Specify the nature and limits of this study as a bioinformatic one. When referring to the scope and objective of the study you should indicate that the validity and credibility of the study and conclusions are greatly dependent on the volume of the available data.
The sentence L82-85 is unclear and confusing. It should be logically be rephrased to express the hypothesis of the current research. Is it logical that HSP20s confer various environmental stresses and at the same time can be developed to resist adverse environmental stress??
The objective and significance of the research should be clearly stated at the end of the introduction section.
§ Minor
L35, change (single codon) to (single codons)
L37, change (a number of) to (several).
L37, change (are not at present) to (is not at present)
L46-47, change (to evolutionary history) to (to the evolutionary history)
L48, change (was proposed) to (was concluded)
L51, change (is promising method) to (is a promising method)
L55, change (genes deriving from) to (genes derived from)
L56, change (in regulation of) to (in the regulation of)
L57, change (participation of) to (the participation of)
L57, change (response to) tom (the response to)
L59-60, change (patternsmight) to (patterns might)
L61, change (heat shock protein 20 (HSP20s) are) to (heat shock protein 20 (HSP20s) is)
L71, change (consisted of (32, 30, 51, and 42) putative) to (consisted of 32, 30, 51, and 42 putative).
L73, change (into in) to (into)
L75, change (might be driving) to (might be a driving)
L80, change (and oxidate stress) to (and oxidative stress)
L80-81, change (AsHSP17 increase) to (AsHSP17 increases)
L83, change (the improving) to (improving)
L83-84, change (be one important) to (be an important).
L84, change (with resistance to) to (resistant to adverse)
L85, change (family as model) to (family a model)
L90, change (stress, easy cultivation) to (stress and easy cultivation)
L91, change (as model) to (as a model)
L93, change (a number of efforts) to (several efforts)
Materials and Methods
Major
The original bioinformatics data on which the study was based should be adequately described, delimited and mapped both temporally, spatially, and volumetricallyز
Minor
L110, change (of remaining) to (of the remaining)
L128, change (near by) to (nearby)
L142, change (play role in CUB) to (play a role in CUB).
L161, change (codons referred) to (codons refer)
L166-167, change into (in a high-expression database is 0.08 greater than that in a low-expression database).
L171, change (use of neighbor-joining) into (use of the neighbor-joining)
Results
Major
The authors should describe the obtained results as delimited by the original available data.
Minor
L192, change (with pI value) to (with a pI value)
L212, change (had the relatively high expression levels.) to (had relatively high expression levels.)
L221, change (B. rapa was considered) to (B. rapa were considered)
L240, change (were situated below) to (most of the genes resided below)
L-253-254, change (pyramid) into (pyrimidine).
L255, change (In case that mutation) to (In case mutation)
L282, change (mutation pressure play) to (mutation pressure plays)
L294, change (exhibited correlation) to (exhibited a correlation)
L298, change (Except the G3s) to (Except for the G3s)
L312, change (with the control condition) into (with the control condition)
L345, change (separated in different) to (separated into different)
L372, change (and majority of) to (and the majority of)
Discussion
Major
The authors should specify the drawn conclusions as delimited by the scope of the original available data.
Minor
L381, change (decreases mutation) to (decreases the mutation)
L389, change (on condition that) to (on the condition that)
L391, change (greenhouse gas) to (greenhouse gases)
L391, change (plant will encounter) to (plants will encounter)
L392, change (especially, such as temperature,) to (especially temperature,)
L398, change (at genomic level) to (at the genomic level)
L399, change (associated with genetic relationship.) to (associated with a genetic relationship.)
L402, change (In present study) to (In the present study)
L403, change into (The number of HSP20s in four species is more…………..)
L403-404, change (is more than those in fungi,) to (is more than in fungi)
L406, change (seed plant) to (seed plants)
L408, change (of whole genome) to (of the whole genome)
L409, change (in classification of) to (in the classification of)
L411, change (In present study) to (In the present study)
L412, change (closer relationship) to (closer relationships)
L415, change (issue was to be addressed) to (issue to be addressed)
L419, change (were AT rich) to (were AT-rich)
L420, change (during evolutionary) to (during the evolutionary)
L425, change (Previous study) to (Previous studies)
L441, change (possibly by) to (probably by)
L450, change (in neutrality plot,) to (in the neutrality plot,)
L455, change (showed close) to (showed a close)
L457, change (shared more similar) to (shared a more similar)
L459, change (at gene level) to (at the gene level)
Conclusions
Major
Specify and delimit the conclusion according to the volume and specification of the original bioinformatic data.
Minor
L464, change (to contribute the) to (to contribute to the)
L466, change (the consistence) to (the consistency)
L467, change (important evidences) to (necessary evidence)
L468, change (in future.) to ( in the future.)
Comments on the Quality of English Language
Moderate editing is needed
Round 2
Reviewer 2 Report
Comments and Suggestions for Authors
Thank you for carefully addressing my comments on version 1
Reviewer 3 Report
Comments and Suggestions for Authors
The authors have responded satisfactorily to all the comments provided.
Comments on the Quality of English LanguageThe English of the article has been improved. Ordinary linguistic editing will be, however, beneficial.